# 'People have started to deliver in the facility these days': a qualitative exploration of factors affecting facility delivery in Ethiopia

Zelee Hill,[1] Yared Amare,[2] Pauline Scheelbeek,[3] Joanna Schellenberg[3]

[1]Institute for Global Health, University College London, London, UK
[2]Consultancy for Social Development, Addis Ababa, Ethiopia
[3]London School of Hygiene and Tropical Medicine, London, UK

**Correspondence to**
Dr Zelee Hill; z.hill@ucl.ac.uk

## ABSTRACT

**Objectives** To understand the recent rise in facility deliveries in Ethiopia.
**Design** A qualitative study.
**Setting** Four rural communities in two regions of Ethiopia.
**Participants** 12 narrative, 12 in-depth interviews and four focus group discussions with recently delivered women; and four focus group discussions with each of grandmothers, fathers and community health workers.
**Results** We found that several interwoven factors led to the increase in facility deliveries, and that respondents reported that the importance of these factors varied over time. The initial catalysts were a saturation of messages around facility delivery, improved accessibility of facilities, the prohibition of traditional birth attendants, and elders having less influence on deciding the place of delivery. Once women started to deliver in facilities, the drivers of the behaviour changed as women had positive experiences. As more women began delivering in facilities, families shared positive experiences of the facilities, leading to others deciding to deliver in a facility.
**Conclusion** Our findings highlight the need to employ strategies that act at multiple levels, and that both push and pull families to health facilities.

## Strengths and limitations of this study

► The use of qualitative methods to understand a complex issue.
► Use of multiple study sites, purposive sampling to saturation, reflexivity, triangulation of methods and respondent groups and within and cross case analysis.
► The study sites were all reasonably accessible, within half an hour walk from a motorable road, and had reasonably functioning health extension worker systems. The study may thus underestimate accessibility issues, and the mechanism related to message saturation is unlikely to be triggered where health extension workers function suboptimally.
► There is the potential for social desirability bias given the pressure for women to deliver in a health facility, and because study mothers were mainly identified by health extension workers or members of the Health Development Army.

## INTRODUCTION

Sub-Saharan Africa accounts for an estimated 66% of all maternal deaths.[1] Despite recent declines, mortality rates are still unacceptably high. Most deaths are preventable and occur around the time of delivery. Although it is essential to address the broad determinants of maternal mortality, such as female education and social status,[2] having a skilled attendant at delivery is nevertheless considered to be the most critical intervention.[3 4]

Ethiopia had one of the lowest levels of facility delivery coverage in the world, with Demographic and Health Survey (DHS) data showing coverage of 5% in 2000, 6% in 2005 and 11% in 2011 (3-year recall).[5] This low coverage persisted despite government efforts to increase the accessibility and availability of services.[6 7] Efforts included the introduction

of the Health Extension Program in 2003, where two female health extension workers (HEW), stationed at newly constructed rural health posts, and paid around $100 a month, were trained for 1 year to serve around 5000 people. They provide health promotion, disease prevention and treatment, and work both at the health post and in the community.[8–11] Their role includes providing care to women through pregnancy, birth and postnatally including providing Antenatal Care (ANC) and promoting birth preparedness and complication readiness.[12] The programme had little impact on the coverage of skilled delivery, even when HEW were trained to conduct deliveries in the health post,[7 12] and studies have found persistent and multiple barriers to change.[13–20] HEWs no longer provide delivery services but rather assist delivering women in reaching health centres staffed by skilled birth attendants.[21]

By 2016 the DHS, using the same methodology to measure coverage as in previous

surveys, found that coverage increased to 33% (3-year recall, with a regional range of 18%–97%), a 200% increase from 2011.[22] The Ethiopian government attributes the increase largely to the work of the Health Development Army (HDA).[23–25] The HDA, created in 2012, is a network of all women in rural areas, organised into development groups of 30 women (1–30 networks), who are further clustered into groups of 6 (1–5 networks).[20 26–28] Groups select a leader who is then trained and supervised by the HEW. The HDA leaders help members adopt practices promoted by the HEW, hold participatory learning and action meetings, link pregnant women with care providers, hold monthly meetings for pregnant women, mobilise communities to contribute resources to make facilities mother friendly and facilitate the use of either traditional or modern ambulances.[11 20 23 27]

Around the same time as the creation of the HDA, there were other policy changes that could have influenced facility delivery rates. These include the prohibition of the use of traditional birth attendants (TBAs) for delivery,[20 26] and the provision of a four-wheel drive ambulance to transport women to facilities for delivery to every rural district.[29] In addition the number of health centres, staffed with two midwives, increased and the road infrastructure was also improved.[26 30] In this paper we use qualitative data to explore the reasons for the increase in facility deliveries in four study sites, we used a phenomenological approach as we were interested in understanding lived experiences.

## METHODS
### Study setting selection and characteristics
Data were collected between March and May 2015, from two wards (*kebeles*), the smallest unit of local government, in the Southern Nations, Nationalities and Peoples (SNNP) region and two in Amhara region. Amhara has shown an increase in facility deliveries from 11% (2011) to 35% (2016), and SNNP region from 7% to 33%.[5]

Data were collected from areas where 'The Last Ten Kilometers' (L10K) programme was active in supporting the Health Extension Program. *Kebeles* were selected from a list, provided by L10K project staff, of *kebeles* considered to have a reasonably functioning HEW system, that is that they had HEWs in place that were considered to be active

and working well. Other selection criteria were that the *kebeles* were seen as typical of the district (*woreda*) with no unusual characteristics such as having a large hospital or a large industry close by, and were less than half an hours walk from a motorable road so that the study team could feasibly access them. We have labelled these *kebele* 'A-D' to maintain anonymity. Table 1 shows the characteristics of the selected *kebele*s, all of which had a subsistence farming based economy. Although the study sites were all a short walk from a motorable road, access to public transport was very limited.

### Data collection
Data were collected as part of a study to understand how HEWs influence maternal and newborn care behaviours, of which facility delivery was one. Four trained interviewers collected data in the local language using pretested semi structured guides developed by the authors. When needed translators were used. The content of the guides was informed by a theoretical framework, which identified pathways through which HEWs could influence behaviours by modifying families capabilities, opportunities and motivation.[31] Data were collected from mothers, grandmothers, fathers, HEWs and HDA leaders using narrative interviews, in-depth interviews (IDIs) and focus group discussions (FGDs). All community respondents had children or grandchildren under 12 months of age, with narrative mothers having children less than 3 months of age to facilitate recall. Using a range of both methods and respondents allowed for data triangulation and ensured we captured a range of viewpoints. Narrative interviews with mothers were used to capture personal experiences, in-depth interviews to capture perceptions of what was commonly done in the community, and focus group discussions to collect data that we felt would benefit from being discussed in a group interaction.

Data were collected until saturation was reached, that is, until additional interviews provided similar information to that already obtained. Saturation was determined by frequent transcript reviews. The sample size, respondent groups and the interview content related to facility delivery are shown in table 2. In the FGD, we employed several activity oriented exercises such as sorting and ranking to encourage group interaction and participation

| Table 1 | Characteristics of study *kebele* | | | | |
|---|---|---|---|---|---|
| **Region** | ***Kebele*** | **Ethnicity** | **Predominant religion** | **Access to health centres** | **Terrain** |
| Amhara | *Kebele* A | Amhara | Orthodox Christian | Moderate | Hilly |
| | *Kebele* B | Amhara | Orthodox Christian | High | Hilly |
| SNNPR | *Kebele* C | Gamo/ Wolaita | Protestant/ Orthodox Christian | High | Predominantly flat with some hilly parts |
| | *Kebele* D | Silte | Muslim | Moderate | Flat |

SNNPR, Southern Nations, Nationalities and Peoples Region.

**Table 2** Data collection method, sample size and content related to facility delivery

| Method | Sample | Interview content related to facility delivery |
|---|---|---|
| Narrative interviews with recent mothers | 12 | ► Labour and delivery story.<br>► Perceived knowledge and skills related to pregnancy and newborn care.<br>► Description of contacts during pregnancy and delivery with health workers, HDA leaders and HEWs.<br>► Information received on where to deliver, source of the information, reaction to the information and the impact of the information on decision-making. |
| In-depth interviews with recent mothers | 13 | ► Perceptions of where most people deliver, and community views of those who deliver at home and those who deliver in a facility.<br>► Views on HEW/HDA leaders work and their suitability.<br>► Description of contacts during pregnancy and delivery with health workers, HDA leaderss and HEWs.<br>► Information received on where to deliver, source of the information and reaction to the information.<br>► Most significant maternal and newborn health changes in the community in the last 2 years, and why things changed. |
| FGD with recent mothers | 4 | ► Pile sort of behaviours practised/not practised, important/not important and that are promoted/not promoted by HEWs/HDA leaders.<br>► Community views of those who deliver at home and those who deliver in a facility.<br>► Most significant maternal and newborn health changes in the last 2 years, and why things changed.<br>► Reaction to statements that HEWs/HDA leaders work does not bring change, and that people dislike HEWs telling them where to deliver. |
| FGD with grand-mothers | 4 | ► Reaction to a picture of a facility delivery.<br>► Most significant maternal and newborn health changes in the last 2 years and why things changed.<br>► Reaction to statements about grandmothers supporting traditional practices, and that mothers do not listen to grandmother advice. |
| FGDs with fathers | 4 | ► Reaction to a picture of a facility delivery.<br>► Fathers role in deciding place of delivery.<br>► Response to a scenario where a family does not follow HEW advice.<br>► Views on HEW/HDA leaders work and their suitability.<br>► Most significant maternal and newborn health change in the last 2 years and why things changed.<br>► Reaction to statements that mothers/fathers make decisions about delivery, and that people dislike HEWs telling them where to deliver. |
| FGD with HEW and HDA leaders | 4 | ► Pile sort of behaviours practised/not practised, important/not important and that are promoted/not promoted by HEWs/HDA leaders.<br>► Most significant changes in the community, and in their work, related to maternal and newborn health in the last 2 years and why things changed.<br>► Successes and challenges they faced in encouraging behaviour change.<br>► Reaction to statements that mothers get punished if they do not follow advice, and that mothers prefer advice of family members. |

HDA, Health Development Army; HEW, health extension workers; FGD, focus group discussion.

and reduce social desirability bias, which can be a particular issue in Ethiopia.[32]

Mothers, grandmothers and fathers, from different households, were identified by the HEW/HDA leaders or through snowball sampling from the community respondents – with the first method providing the majority of respondents. Eligibility criteria were that the family had received at least one visit by an HEW or HDA leader. Mothers were selected to ensure diversity in age, educational level, parity, sex of newborn and socioeconomic status. Grandmothers could be paternal or maternal—dependent on which was closest to the family. We also aimed to get diversity in place of delivery, but located few women who admitted they delivered at home. All of the HDA leaders in the study *kebeles* were invited for the HDA FGDs. As there were only two HEWs per *kebele,* HEW FGDs included HEWs from neighbouring *kebeles.* Interviewers approached potential respondents in their home, or at the health post. Three respondents refused, as they were too busy. Interviews lasted from 1 to 2 hours and took place in respondents' houses, or the health post for the HEW. FGDs were conducted with 3–7 *respondents* in neutral locations and lasted from 1.5 to 2.5 hours. HEWs and HDA leaders were not present

during any of the interviews or FGDs with community members.

Interviews and FGDs were audio-recorded and fully transcribed by the data collectors in English as soon as possible. Data collectors met regularly during fieldwork to discuss emerging themes and to receive feedback from the senior researchers. On entering a householdthe interviewer introduced themselves and the project to key people, and gave the head of household a project leaflet. They explained who they wanted to interview and read aloud a study information sheet to them in a quiet place. For FGDs the information was read aloud to all FGD respondents. The interviewers checked respondents' comprehension, rephrased if necessary and gave the respondents an opportunity to ask questions. If the respondent agreed to be interviewed the interviewer read the consent form out loud and asked the respondent to sign to show that they were willing to be interviewed, understood the study, were happy for their words to be written down and recorded, were happy for their quotes to be used and for the information collected to be transferred to London. The interviewers also signed each form.

### Respondent and public involvement
Respondents were not directly involved in the design of the study, however the interview guides were iterative and were modified as the research progressed based on reported experiences and perceptions. Some respondents were recruited through snowball sampling, that is, where respondents suggested others they knew who were eligible for interview.

### Data analysis
Analysis began during data collection through regular team meetings and reflection. A formal analysis session was held with the data collectors in the middle and at the end of data collection, this included discussion of how our characteristics could have influenced how data were collected and interpreted. Once data were collected all transcripts were read several times to ensure familiarity with the data, to begin to identify notable constructs, and to see the data as a whole. A deductive coding template was developed in Nvivo based on the theoretical framework that guided the interview content. Interviews and focus groups were then coded inductively within these broad themes. Coding was done by identifying the underlying meaning of each section of text and how it was different or similar to others section. Codes that contained similar concepts were then put into larger themes. Themes and codes were modified by looking for patterns, links and contradictions within themes. Data credibility was checked by triangulating data between respondent groups and between data collection methods. Data analysis was done by three of the senior researchers, who discussed their coding regularly to enhance conceptual thinking and to increase coding rigour. Reflective notes were kept throughout the process.

**Table 3** Sample characteristics (narrative and mother IDIs)

| Characteristic | Frequency (n=25) |
|---|---|
| Age | |
| ≤ 24 | 10 |
| 25 – 34 | 10 |
| ≥ 35 | 5 |
| Education | |
| None | 10 |
| Primary | 12 |
| Secondary and above | 3 |
| Religion | |
| Islamic | 8 |
| Christian | 17 |
| Parity | |
| 1 | 7 |
| 2–3 | 7 |
| ≥4 | 11 |
| Place of last delivery | |
| Home | 6 |
| Facility | 19 |

### RESULTS
Table 3 shows the characteristics of the narrative and IDI respondents. Respondents had a range of ages, education levels, parities and religion. We did not achieve the planned diversity in place of delivery, as 19/25 of the narrative/IDI women had delivered in a facility. This is possibly because families were reluctant to admit to home deliveries, and because facility delivery rates may have been high in the study area because the sites were relatively accessible, within walking distance of a motorable road, and had functioning HEW systems. In addition, HEW/HDA leaders assisted in identifying respondents, and may have favoured those who delivered in a facility.

The FGD participant mothers were varied in age (range 19–35 years of age), parity (range 1–7 children), education (none-secondary level) and ethnicity. The FGD participant fathers were older (range 28–45 years of age), and the FGD participant grandmothers were less educated with almost all being uneducated. Grandmothers were predominantly, but not exclusively, paternal.

All respondent groups reported that the increase in facility delivery was recent, and that previous attempts to encourage facility delivery had limited success:

> People have started to deliver in the facility these days… They [families] used to give us lots of excuses like, let the cattle return back home, let the sun start setting, and let's wait for this and that; believing that the mother would deliver in the meantime…so that used to be very problematic [HEW FGD, kebele A Amhara].

At the time of data collection, delivering in a health centre was reported as the usual practice in all study communities, and respondents reported that: 'everyone knows what to do'… 'no one delivers at home'.

Respondents reported that the factors that influenced facility delivery changed over time and consisted of push and pull factors. We identified four themes around the initial uptake of facility delivery: saturation of messages around facility delivery, improved accessibility of facilities, the prohibition of TBAs and elders having less influence on deciding the place of delivery. Themes around the drivers of facility delivery once uptake had begun were around families having positive experiences of facilities, seeing the worth of a facility delivery and sharing their positive experiences with others.

### Saturation of messages and contacts

All mothers interviewed reported receiving information on the importance of facility delivery, and all respondent groups knew that it was being strongly promoted at: 'every opportunity'. In the majority of cases mothers had received information from at least two sources, and at several time points:

> [HDA] tells me about it repeatedly and forcefully [Mother IDI, kebele D SNNPR].

Information was mainly given at the health post, at home, during ANC and at community meetings. The main sources of information were the HEWs and HDA leaders. The resultant high awareness levels was reported as a reason for the increase in facility delivery rates:

> Interviewer: Why didn't you go [to the facility] at that time [for previous deliveries]?
>
> Respondent: Because there was no one who educates you like this at that time. Nobody advised us to deliver in the health center… My knowledge was not as strong at that time [Mother IDI, kebele D SNNPR].

A theme that emerged from all respondent groups, related to how the information on facility delivery was received, was around community trust in health workers and HEWs. Trust in HEWs arose from a view that HEWs were knowledgeable because of their training and were higher status than community members:

> 'They [HEW] are better than us; they teach us what they have learned. She [HEW] went there [training] so that she could bring us some good education, we don't believe she teaches us harmful advice' [Mother FGD, kebele D SNNPR].

The theme around trust in HEWs was contrasted by views of the HDA leaders who were less trusted as they were viewed as people who transferred messages rather than being knowledgeable in their own right. But, HDA leaders played a key role in ensuring the penetration of messages, and by informing HEWs about pregnant women:

> The leaders of this group [1–5 HDA group] follow how many of them are pregnant… The leaders know everything about their group… And when labor starts, the leader will inform the HEWs [Mother IDI, kebele B Amhara].

### Improved accessibility

Knowledge of the ambulance service was universal across respondent groups. In some sites families were given the ambulance number during pregnancy. In other sites families called the HEW at the start of labour, and the HEWs then called the ambulance. The presence of the ambulance was reported as facilitating facility delivery by all groups:

> If ambulance service had not started functioning in the kebele, the mother surely gives birth at home [Mother FGD, kebele A Amhara].

> This time there is no one who delivered at home, it was in our fathers' time, now there is ambulance which take the mother to the health center, so all women deliver there [Mother FGD, kebele A Amhara].

A theme among fathers was the role that the increase in the number of health centres, free delivery care and the construction of roads had on facility delivery uptake:

> Formerly people think there is a payment for delivery, like they pay for treatment but there is no such things… there were a lot of people who deliver at home thinking it [facility delivery] needs money… even if she (mother) asked to go the husband didn't want, thinking he will be asked for money [Father FGD, kebele D SNNPR].

Despite the reported importance of ambulances, several families interviewed described a problem accessing the service. This was most frequently because the ambulance was busy, could not come because of heavy rain, had no fuel or took too long to come:

> I asked [HEW] to call an ambulance, but there was no ambulance so we were told to use public transport [Grandmother FGD, kebele B Amhara].

In the narrative interviews over half of the women (6/9) who called an ambulance had a problem accessing the service, of these three delivered at home or with the HEW, one delivered in the health centre but waited a long time for the ambulance and the other two took public transport.

The four study sites were reasonably accessible and respondents talked of villages where health facility delivery was still very difficult because of accessibility issues:

> Not all villages are accessible, those who live in X village, they are not able to deliver in health facility. But those who live in nearer villages… It is a must to deliver in health facility since they are close [Mother IDI, kebele A Amhara].

## Prohibition of TBAs

In all sites the use of TBAs was reported as forbidden, with a threat of sanctions for those who conducted or had a home delivery:

> During pregnancy, they [HEW] told me not to deliver at home. They said 'if you deliver at home, you will be punished'… 500 birr [$22] if I deliver at home [Mother IDI, kebele C SNNPR].

> …birth attendants are not willing to assist due to fear of punishment… the women herself will pay 2000 birr [$88] and the birth attendant will pay 1000 birr [$44] [Mother IDI, kebele B Amhara].

Decisions about sanctions were made at community level through the 1–5 or 1–30 HDA networks:

> The community decided a 'Sera' [customary law] that if a mother delivers at home she will be fined 500 birr [$22] [Narrative interview, kebele B Amhara].

The reported sanctions were varied, with the fines ranging from 200 to 2000 birr [$9–$88].

The possibility of being sanctioned for delivering at home was a key theme relating to the initial uptake of messages around facility deliveries among all respondent groups:

> Interviewer: What do you think brought this change?

> Respondent:… the fear of the punishment, I don't think most of the community understood the benefit of delivering in the facility… didn't give due attention to the lessons… it is after we are told we will be punished [Mother IDI, kebele C SNNPR].

> Respondent 2: There is 1 to 5 [HDA group], and one watch over the other, and there is also punishment; if the mother deliver at home she will be fined with 500 birr [$22]

> Respondent 3: There is a law they are fined… after that people start saying hurry up please she is going to deliver [laughing] [Father FGD, kebele C SNNPR].

Respondents, including former TBAs, reported that the sanctions were justified and beneficial as times were changing. HEWs were aware of them, and at times encouraged their use:

> Respondent: When a mother delivers at home and if the baby dies…the government will prosecute her for that… We use such threats

> Interviewer: Who tells them such threats?

> Respondent: We call the HDA leaders (1 to 30) and then tell them that such threats may work, and then they go and tell the mothers [HEW FGD, kebele C SNNPR].

In only one site were there reports that sanctions had been used in practice, and the respondents that we interviewed who had delivered at home reported that they had been excused the sanctions:

> [HEW said] If you were another person I will take you to jail but you face a lot of problem that's why I left you [IDI mother, kebele D SNNPR].

Although the respondents who delivered at home reported that sanctions were not applied, they did report that the HEWs were angry with them and, in a few cases, denied them services:

> She [HEW] suspected that I hid and delivered at home… and was very angry. Because she was angry then, she did not tell me things…there was no mention of how I should be bathing the baby and the like [Narrative woman, kebele B Amhara].

## Power shift

Grandmothers had little influence on place of delivery, and respondents in all groups used words like: 'we are in a different time' and 'time has changed'. This lack of influence was attributed to mothers being modern because of the education given by the HEW, were thus more knowledgeable than their elders and were consequently able to challenge their advice:

> Today's mothers are young and modern. They easily accept new ideas… they wouldn't like to do the traditional practice… since they have received the new education [Narrative mother, kebele C SNNPR].

Husbands were viewed as having the ultimate decision-making power in the household and generally supported facility delivery. This support from husbands put mothers in a stronger position if they faced opposition from their elders. We found very few grandmothers who reported that they were resistant to the change in delivery location.

## Positive experiences

The main theme around how the drivers of facility delivery changed over time was around the influence of families having and sharing positive experiences of facility delivery:

> I felt very happy [to deliver in the facility]… The doctors give morale, they said take it easy, be strong and the like' [Mother IDI, kebele D SNNPR].

> But now, it is not that they (families) are afraid of the punishment, they have started saying that they are going because they want to get care from the health professionals… They have started saying that the physicians do all they can and help them deliver [HDA FGD, kebele D SNNPR].

> Formerly they feared the penalty, but now those who delivered there (at health center) talk about good thing of delivering there [Mother IDI, kebele C SNNPR].

All respondents in all groups reported that facility delivery was safer and reduced deaths. The provision of an injection to stop bleeding was the most frequently

mentioned benefit along with getting a vaccination for the baby, removal of dirt from the abdomen/stomach, the baby being cared for and not left alone, the facility being hygienic, delivery being less painful and more predictable, and that the facility could deal with problems such as the baby being in the wrong position, the placenta getting stuck or the baby being born weak:

> I know that I will not be hurt if I deliver in the health center… they will inject me something which stopped the excessive bleeding; I know that they will assist me if I will have any complications [Mother IDI, kebele C SNNPR].

> Here (home) there is only suffering until the delivery nothing else, and we are delivering in the facility in a very relaxed way… they [health workers] measure and tell us how much time is left…but here we don't know anything, we are just laboring and waiting until we deliver or die [Mother IDI, kebele D SNNPR].

All respondent groups reported that at the facility mothers received food, drink and sometimes a cloth for the baby:

> Everything is perfect, even porridge and gruel is prepared and served to mothers in the health center. It is really good… Even the person who accompanied them is invited [Grandmother FGD, kebele B Amhara].

A theme related to grandmothers was that becasue they were allowed in the delivery room they had less fears about what occurred during a facility delivery:

> We (health workers) never used to allow anybody inside the delivery… as they [family members] start to see; they started saying 'we were afraid that you would insert materials inside her'… they see that things are good [HEW FGD, kebele A Amhara].

Not only did respondents report benefits and good experiences of delivering at the facility, they often shared their experience with their friends and neighbours, and several mothers reported that their friends and neighbours had influenced their delivery location:

> I heard from other people, I heard that it is good to deliver in the health center, so I was planning to deliver there… one of my neighbors delivered in the health facility, and she told me it is good… She told me the mother will be very clean, she will not have bleeding… health professionals would help the mother and save her life. [Mother narrative, kebele C SNNPR].

> Respondent 1: Now, the unwilling ones also went there because she saw when others do

> Respondent 2: Yes, formerly, they did not want to be exposed. They said that, Saint Mary will do what she wants. But now, they see the benefit. And learn one from the other' [HDA FGD, kebele B Amhara].

## DISCUSSION

A systematic review of qualitative studies exploring facility delivery classified the findings based on the quality and coherence of studies.[33] Barriers to facility delivery in which there was high confidence were: cultural barriers, such as perceptions of birth as a natural event; decision-making barriers, including the role of elder women; proximity, access and cost barriers; a reliance on TBAs; and barriers related to perceived poor quality of care and mistreatment by health workers. High confidence facilitators were valuing facilities for complications and perceiving them as providing high quality of care. Previous birth experiences were both a barrier and a facilitator. Previous studies specific to Ethiopia identified similar barriers.[13–20] In our study none of these barriers were reported, with the exception of accessibility issues for more remote villages. Our findings suggest that these barriers were overcome through a combination of saturation of messages around facility delivery from trusted sources, reduction in access issues, the prohibition of TBAs, power shifts away from grandmothers and positive experiences. The focus of this paper on what has driven the change process adds new insights to the literature, which to date has focused on barriers and facilitators to uptake rather than mechanisms of change. It is widely recognised that comprehensive efforts, at multiple levels, are required to successfully increase facility delivery rates,[33] this is what has occurred in the study sites. Previous interventions in Ethiopia that have focused on access barriers at one level have not been successful.[30]

Respondents reported that the drivers of behaviour change in our study sites varied over time. One of the initial catalysts was the prohibition of TBAs. TBAs have been prohibited in several other African countries, but the policy has often encountered problems such as the ban being ineffectual due to enforcement issues, TBAs continuing their work underground, accessibility remaining a key barrier and poor quality facilities limiting the effects.[34–40] Our data suggest that, in our study sites, the ban has been effective. This could be because the ban was coupled with increased awareness of, and access to, alternative options that were viewed positively and because the HDA model allowing pregnant women to be identified and followed. We were unable to locate details of the TBA ban, but our data suggest that the specifics of the ban, in relation to whether and how it was implemented, were determined at local levels. There may be considerable variation in implementation and impact in other Ethiopian settings.

Prohibiting home births is controversial, and it has been argued that it infringes on personal choice and autonomy.[36 37] This is exemplified in a ruling in the European Court of Human Rights that regulations which make home births difficult to obtain violate the right to a private life.[41 42] On the other hand it has been argued that the restrictions implemented in several African countries are made in the interest of public health and are thus justified. Whatever view is taken, an important

consideration is whether such bans and sanctions result in families hiding home deliveries, and subsequently reducing their care seeking. For example, in Burkina Faso sanctions and verbal abuse for those who did not attend services resulted in these families being too fearful to access services when they needed them.[43]

In our study as families began to experience facility deliveries, the driver changed from push factors to the pull of a desire for facility deliveries. Positive experiences at the facility changed perceptions, and neighbours shared their experiences. If families continue to have and share positive experiences the increase is likely to be sustained.

Saturation of messages was a key driver in increasing awareness and uptake of facility deliveries. The importance of achieving saturation of messages is often overlooked within public health behaviour change, and achieving high exposure has received less attention than the development of high quality messages, yet high exposure appears to be equally important for success.[44 45] The HEW and HDA model, which was functional in our study sites, facilitated high exposure as pregnant women could be identified and followed.

We found that elders had lost their decision-making power because their sons and daughters, due to information from the HEW, were now perceived as being more knowledgeable. There is often an assumption that elders are resistant to change, but we found, as others have,[46] that their views can change rapidly in some circumstances.

We found that different respondent groups highlighted different reasons for change. Efforts to make facilities friendly such as allowing family in the delivery room were particularly appreciated by grandmothers while husbands, who have been identified as making financial decisions regarding delivery location in other studies,[47 48] were the only group to report on the impact of reduced costs, improved roads and a greater number of health centres. Collecting data from multiple groups is important both for the design and evaluation of interventions.

### Data quality and study limitations

We took several steps to maximise data quality, and took measures to improve the transferability of our findings including: using multiple study sites, purposive sampling to saturation, reflexivity, triangulation of methods and respondent groups and within and cross case analysis.[49 50] Despite this the findings may not apply to other areas with significantly different contextual issues. For example the study sites were all reasonably accessible and had reasonably functioning HEW systems. It is likely that distance and accessibility are the main factors influencing delivery location in less accessible areas, with our respondents reporting that they knew of areas where women were unable to deliver in facilities because of distance. Studies in other settings in Ethiopia would further enhance transferability, however, the study findings suggest several issues that could be considered when exploring issues related to facility delivery coverage and the effectiveness

of interventions to increase facility delivery rates in other settings.

The main limitation of our data is the potential for social desirability bias given the pressure for women to deliver in a health facility—highlighted by the difficulty we had identifying women who delivered at home. The potential for social desirability bias has been identified as particularly high in Ethiopia given a political context that may limit how freely respondents feel able to speak.[32] Although we used methods to help overcome such bias, respondents may still have been unwilling to say negative things about facility delivery, especially those identified for interview by the HEW/HDA leaders. In addition, those respondents identified by the HEW/HDA leaders may have been selected because of their positive attitudes and experiences. To try to reduce this we used snowball sampling to identify respondents, but the majority of the respondents were identified through the HEWs/HDA leaders. As a result, the study respondents may have had different attitudes and experiences to families that were less favoured by the HEW/HDA leaders.

Our findings highlight the need to employ strategies that act at multiple levels, and that both push and pull families to health facilities. The ability to achieve saturation and penetration of messages and to identify and follow pregnant women was a key factor in increasing facility deliveries, this is likely to have been influenced by the unique administrative and political context of Ethiopia. The increase is likely to be sustained if families' experiences of health facilities continue to be positive and effort to improve the accessibility and quality of care continue; such as the provision and maintenance of ambulances, allowing family and cultural ceremonies into the delivery room, and the provision of food at the facility. Given the unique context it is difficult to transfer findings to other countries where, for example, an HDA type network may not function as well. But, we feel that the key messages of focusing interventions at multiple levels, addressing pull and push factors and ensuring saturation of messages are useful for policy makers in other settings to consider.

**Contributors** ZH and JS conceived the study. ZH, YA and PS designed the data collection tools and strategy, YA and PS oversaw data collection, ZH, AY and PS conducted the analysis. ZH wrote the first draft of the manuscript and all authors contributed to the re-drafting and revision of the manuscript.

**Funding** This work was supported by the Bill and Melinda Gates Foundation as part of the IDEAS study Grant number 0PP1017031.

**Competing interests** None declared.

**Patient consent for publication** Not required.

**Ethics approval** Ethical approval was granted by the Ministry of Science and Technology in Ethiopia and the London School of Hygiene and Tropical Medicine in the UK.

**Provenance and peer review** Not commissioned; externally peer reviewed.

**Data sharing statement** Unpublished data from this study are not available. Data anonymity is not obtainable given the depth of the narrative qualitative transcripts, even when these have been anonymised.

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
