## [Reviewer comments · BMJ Open]

ARTICLE DETAILS

TITLE (PROVISIONAL)	"People have started to deliver in the facility these days": A qualitative exploration of factors affecting facility delivery in Ethiopia
AUTHORS	Hill, Z; Amare, Yared; Scheelbeek, Pauline; Schellenberg, Joanna

VERSION 1 - REVIEW

REVIEWER	Elizabeth Kaselitz University of Michigan, United States
REVIEW RETURNED	27-Aug-2018

GENERAL COMMENTS	This manuscript provides a meaningful contribution to the literature and our understanding of what leads women to deliver in facilities. I appreciate the discussion of push vs. pull tactics and the importance of employing multiple methods at once to reach different populations and family members with differing interests - which makes sense given the astonishing 200% increase in delivery rates over such a short period of time. The only thing lacking for me in the manuscript was addressing whether there were any changes in reporting methods, or incentives for better reporting, or any other reasons why the data on facility deliveries may have been artificially inflated in recent years, or under-reported in the past. Or just a statement verifying that there have been no changes in how these data were collected or incentives for better reporting over this time period would suffice. That is just where my mind initially went - to changes in reporting potentially accounting for some of that increase - and a statement quelling any concerns about this would add to the paper for me. Lines 3 - lines 41-43- needs editing. Page 6 – line 27 – based “on” Page 6 – line 34 – discussion “of” Page 11 lines 37-38– development “of high-quality” messages Page 12 – line 37 – “unique” context Overall, strong contribution to the literature, interesting read, and my recommendation is for this manuscript to be accepted.
--

REVIEWER	Sarah Rudrum Acadia University Canada
REVIEW RETURNED	03-Sep-2018

GENERAL COMMENTS	I enjoyed having the opportunity to review the article. I was interested in some of the methodological choices, such as photo elicitation & responding to statements. My review is attached. Summary & Contribution The research focuses on the reasons for an uptick in facility-based deliveries in Ethiopia, using interview and focus group data. Participants included mothers, other family members, and health workers. This area of study is important because the current international approach to improving maternal health and reducing maternal mortality prioritizes increasing facility-based deliveries. The methods used are rigorous and innovative, and help to understand the complexity of policy, health provision, and family/community factors influencing birth location. The authors find that multiple levels of intervention are necessary, that both push and pull factors are at play, and that saturation of messages is important. Substantive suggested revisions & questions  1. The data collection occurred during a short period in 2015, yet major findings claim a result of change over time: “We found the factors that influenced facility delivery changed over time and consisted of push and pull factors.” (and again on p. 11 “We found that the drivers of behavior change in our study sites varied over time”)The authors should consider revising to this to state that the factors were reported to change over time, or that participants identified that the factors changes over time, as the current presentation of findings seems to suggest a comparative or longitudinal design. 2. On p. 12, around line 11, it seems a new heading is needed, as this is no longer discussion 3. I’m not sure about the claim that saturation of messages is more important than quality of health message. (Logically, saturation of a wrong message would be problematic.) Perhaps revise to also important, or equally important? (This is p. 11) 4. The efforts to improve quality of care might be underplayed here. On p. 12, efforts to include family in the delivery room, offer food, reduce cost, and otherwise make care more accessible and culturally safe are discussed. This seems significant and could be highlighted in the conclusion/ overall assessment of drivers of change. (A contrast is settings where there is message saturation but facility care remains fairly poor or hostile.) 5. The introduction states: “Although it is essential to address the broad determinants of maternal mortality, such as female education and social status [2], having a skilled attendant at delivery is still considered to be the most critical intervention [3 4].” I’m wondering if “nevertheless” would be a better word choice than “still” here, because the focus on facility based delivery is to some extent a departure from previous efforts (such as to train TBAs).
---

	Minor suggested edits  1. suggest changing the word “that” to “who” on p. 6 line 30 2. comma splice, page 11 line 5 and 6
--	--

REVIEWER	Barbara Madaj Liverpool School of Tropical Medicine
REVIEW RETURNED	25-Oct-2018

GENERAL COMMENTS	The study presented covers a very important and relevant topic and therefore in principle merits a publication. However, the current version of the paper requires substantial review before it is ready for publication. In particular:  1. Background but also Results/Discussion: although the sections cover relevant information, at times authors seem to assume knowledge of the area and details of the Ethiopian context, which are not commonly known and therefore require explaining: characteristics of the specific locations of the research sites vis-à-vis facility delivery coverage: the background section suggests on average 33% coverage in 2016 – though with regional differences, so when the authors report challenges of finding women who delivered at home, is that because of the cited fear of repercussions or are the areas actually some of the more ones with higher coverage rates and therefore? What re the 1-5 or 1-30 groups/networks? What is a ‘woreda’? What is a ‘kebele’? Could the authors also comment on aspects such as availability of public transport as alternative means when ambulances are not available to help to explain that even though lack of ambulances was one of the key barriers mentioned, facility-based deliveries seem to be the norm according to the study? Banning of TBA support needs to be explained in more detail to help to contextualise the findings reported in the paper. Similarly, the roles of the CWS needs to be better explained – some information is provided already but it may not be sufficient for readers to understand their role and especially the recent changes which then help to explain the findings of the research presented. 2. Design and methodology require more detailed information included in the paper to allow readers to appreciate the approach and its strengths as well as limitations. Much of the information is covered in broad terms only and therefore does not allow for the in-depth understanding. In particular,  - Details of ethics review and consenting process need to be specified. Also, relation of the researchers (especially in terms of the links with the wider study within which this research is based) needs to be specified - Characteristics of the respondents need to be presented in more detail – at present the text states that recruitment was done in a way to allow for breadth of characteristics but no detail is provided in terms the outcome of the strategy - The justification for the choice of methodology and the execution of the study need to be explained more clearly; otherwise it is difficult to assess the robustness of the data and therefore the findings. Please explain why the qualitative methods applied were
---

selected and what the difference between the different types of data collection methods (in-depth interviews, narrative interviews and focus group discussions) was and why all the were deployed. Regarding respondents, more detail on how they were selected is required especially in light of the acknowledged limitation of potential bias stemming from the recruitment and pool of people involved in the selection. The breaths of respondents – reported as a recruitment strategy – requires substantiating; would the mothers, fathers and grandmothers present the same households? If so, that considerably limits the breadth of responses and needs to be acknowledged; similarly, would snowballing be done via health providers who were also respondents? That again shrinks the pool of responses voiced by those who participated in the research. Overall, it would be helpful to know how many stems for the recruitment (with for gate keepers and snowballing) were used and in case that number is low, to acknowledge and explain that under limitations.

- Definitions of terms such as ‘recent mothers’ (which presumably also expected to the fathers and grandmothers), ‘reasonably accessible’ and ‘reasonably functional’, ‘no unusual characteristics’ need to be specified.

- Are grandmothers the mothers of the mother or the father? Would it make any difference within the cultural context studied?

3. Results:

- Some of the findings seem to lack depth and offer only a limited insight into the situation, even if the conclusions appear to suggest those insights were found in the data – please amend that to allow the readers to benefit from the work done. This may link to the next point on improving the structure of the findings reporting.

- Organisation of the results section: at present the results are presented in a way which makes it difficult to follow; even though main headings are used, the flow seems quite unstructured and therefore should be tightened up and reorganised to provide a clearer narrative which is easier to understand. Applying a similar structure for the individual headers such as enables and then challenges, presenting the perspectives of the different groups of respondents or other differences observed in the study to help to bring out the necessary nuances which will make the study presentations stronger and more informative. One way would be to make the key statements then supported by evidence rather than leaving the narrative to flow and lead to a conclusion which is at times mentioned and at other times implied only.

- More specifically, for the challenges identified or in fact encountered by the respondent, would the authors have any evidence on how they were overcome or could be overcome according to the respondents? As mentioned above with regard to transport, lack of ambulances was noted as a key challenge, yet all the respondents represented families where babies were delivered in facilities, so a method for overcoming the barrier must have been found.

4. Discussion: Useful insights are presented, though relatively limited reflection in terms of placing the current research in the context of the wider body of knowledge – what does the research offer that is unique and adds to the body of knowledge? How much

	of the work is generalisable? What are the next steps? Also, some points raised in the discussion do not appear in the Results – please amend that. 5. Style: there are a number of issues relating to the style of the paper which needs revising:  - Parts of the paper seem to read as too informal, especially the summary table of the topic guide context, as well as the methods section – please revise. - Punctuation needs to be standardised/revised: where quotes are used there is no punctuation (such as a colon) to introduce it and the only way of identifying a quote is by the text appearing in quotation marks and in Italics – this is not sufficient. - Use of tenses should be standardised – either resent or past/reported speech to be applied to the results. - Some of the quotes are not grammatically correct and may benefit from editing – ensuring the meaning is not affected. - Also, at times quotes are not self-explanatory and therefore more context is required (e.g. p. 8 ‘they told me not to deliver at home – who are ‘they’?’). - Language revision suggestion: statement on p. 11 ‘their views can change given the right circumstances’ – use of the word ‘right’ gives the sentence a normative tone and should be revised to sound more neutral. - Bibliography – please review to ensure spellings and use of low/upper case letters is consistent
--	--

VERSION 1 – AUTHOR RESPONSE

Reviewer: Elizabeth Kaselitz

Reviewers comment	Response
The only thing lacking for me in the manuscript was addressing whether there were any changes in reporting methods, or incentives for better reporting, or any other reasons why the data on facility deliveries may have been artificially inflated in recent years, or under-reported in the past. Or just a statement verifying that there have been no changes in how these data were collected or incentives for better reporting over this time period would suffice. That is just where my mind initially went - to changes in reporting potentially accounting for some of that increase - and a statement quelling any concerns about this would add to the paper for me.	The figures cited are from the Demographic and Health Surveys, using their standard module, with no change in how data have been collected over time. To clarify this we now start paragraph 3 on page 3 as follows: ‘By 2016 the DHS, using the same methodology to measure coverage as in previous surveys, show that coverage increased to 33% (3 year recall, with a regional range of 18-97%), a 200% increase from 2011 [21]’.
Lines 3 - lines 41-43- needs editing.	The word ‘we’ has been inserted.
Page 6 – line 27 – based “on”	Change made (second to last paragraph page 7)

Page 6 – line 34 – discussion “of”	Change made (last paragraph page 7)
Page 11 lines 37-38– development “of high-quality” messages	Change made (second paragraph on page 15)
Page 12 – line 37 – “unique” context	Change made (final paragraph page 16)

Reviewer: Sarah Rudrum

Reviewers comment	Response
The data collection occurred during a short period in 2015, yet major findings claim a result of change over time: “We found the factors that influenced facility delivery changed over time and consisted of push and pull factors.” (and again on p. 11 “We found that the drivers of behavior change in our study sites varied over time”). The authors should consider revising to this to state that the factors were reported to change over time, or that participants identified that the factors changes over time, as the current presentation of findings seems to suggest a comparative or longitudinal design	The term ‘participants reported’ has been added to the sentence about changes over time in the abstract, results and discussion.
On p. 12, around line 11, it seems a new heading is needed, as this is no longer discussion	The section now starts with a heading ‘Data quality and study limitations’
I’m not sure about the claim that saturation of messages is more important than quality of health message. (Logically, saturation of a wrong message would be problematic.) Perhaps revise to also important, or equally important? (This is p. 11)	The term ‘more’ has be replaced with ‘equally’.
The efforts to improve quality of care might be underplayed here. On p. 12, efforts to include family in the delivery room, offer food, reduce cost, and otherwise make care more accessible and culturally safe are discussed. This seems significant and could be highlighted in the conclusion/ overall assessment of drivers of change. (A contrast is settings where there is message saturation but facility care remains fairly poor or hostile.)	The following has now been added to the concluding paragraph on page 16: ‘The increase is likely to be sustained if families’ experiences of health facilities continue to be positive and effort to improve the accessibility and quality of care continue; such as the provision and maintenance of ambulances, allowing family and cultural ceremonies into the delivery room, and the provision at the facility.
The introduction states: “Although it is essential to address the broad determinants of maternal mortality, such as female education and social status [2], having a skilled attendant at delivery is still considered to be the most critical intervention [3 4].” I’m wondering if “nevertheless” would be a better word choice than “still” here, because the focus on facility based delivery is to some extent a departure from previous efforts (such as to train TBAs).	‘Still’ has been replaced with ‘nevertheless’.
suggest changing the word “that” to “who” on p. 6 line 30	‘That’ has been replaced with ‘who’.
comma splice, page 11 line 5 and 6	A comma has been inserted and the sentence now reads (second paragraph on page 14):

	‘It is recognized that comprehensive efforts, at multiple levels, are required to successfully increase facility delivery rates’
--	--

Reviewer: Barbara Madaj

The background section suggests on average 33% coverage in 2016 – though with regional differences, so when the authors report challenges of finding women who delivered at home, is that because of the cited fear of repercussions or are the areas actually some of the more ones with higher coverage rates and therefore?	The following has been added at the start of the results in page 9 to help the reader understand why we did not achieve the expected diversity in place of delivery: ‘We did not achieve the planned diversity in place of delivery, as 19/25 of the narrative/IDI women had delivered in a facility. This is possibly because families were reluctant to admit to home deliveries, and because facility delivery rates may have been high in the study area because the sites were relatively accessible, within walking distance of a motorable road, and had functioning HEW systems’.
What re the 1-5 or 1-30 groups/networks? What is a ‘woreda’? What is a ‘kebele’?	The 1-5 and 1-30 networks are explained in paragraph 3 of the introduction: ‘The HDA, created in 2012, is a network of all women in rural areas, organized into development groups of 30 women (1-30 networks), who are further clustered into groups of 6 (1-5 networks) [20 25-27]. Groups select a leader who is then trained and supervised by the HEW. The HDA leaders help members adopt practices promoted by the HEW, hold participatory learning and action meetings, link pregnant women with care providers, hold monthly meetings for pregnant women, mobilize communities to contribute resources to make facilities mother friendly, and ensure the use of either traditional or modern ambulances [11 20 22 26]’. The first paragraph of the methods now includes a definition of Kebele: ‘Data were collected between March and May 2015, from two wards (Kebeles), the smallest unit of local government, in the Southern Nations, Nationalities and Peoples region (SNNP) and two in Amhara region’. Woreda are now defined at first mention (second paragraph of the methods) as ‘districts’.

Could the authors also comment on aspects such as availability of public transport as alternative means when ambulances are not available to help to explain that even though lack of ambulances was one of the key barriers mentioned, facility-based deliveries seem to be the norm according to the study?	We have included, in the second paragraph of page 4, the following in relation to the availability of public transport: 'Although the study sites were a short walk from a motorable access to public transport was very limited'. In the first paragraph on page 11, we have added an explanation of what happened to the narrative women who had problems accessing the ambulance: 'In the narrative interviews over half of the women (6/9) who called an ambulance had a problem accessing the service, of these 3 delivered at home or with the HEW, 1 delivered in the health center but waited a long time for the ambulance and the other 2 took public transport'.
Banning of TBA support needs to be explained in more detail to help to contextualise the findings reported in the paper.	It is difficult to determine the details of the ban. To reflect this we have added the following in the first paragraph of the discussion 'We were unable to locate details of the TBA ban, but our data suggest that the specifics of the ban, in relation to whether and how it was implemented, were determined at local levels. There may be considerable variation in implementation and impact in other Ethiopian settings'.
Similarly, the roles of the CWS needs to be better explained – some information is provided already but it may not be sufficient for readers to understand their role and especially the recent changes which then help to explain the findings of the research presented.	We are unclear what 'CWS' refers to.
Details of ethics review and consenting process need to be specified.	The ethics review and consenting section on page 7 has been expanded and now reads: 'Ethical approval was granted by the Ministry of Science and Technology in Ethiopia and the London School of Hygiene and Tropical Medicine in the UK. Written informed consent was obtained from all respondents. On entering a household the interviewer introduced themselves and the project to key people, and gave the head of household a project leaflet. They explained who they wanted to interview read aloud a study information sheet to them in a quiet place. For FGDs the information was read aloud to all FGD respondents. The interviewers checked respondents' comprehension, rephrased if necessary and gave the respondents an opportunity to ask questions. If the respondent

	agreed to be interviewed the interviewer read the consent form out loud and asked the respondent to sign to show that they were willing to be interviewed, understood the study, were happy for their words to be written down and recorded, were happy for their quotes to be used, and for the information collected to be transferred to London. The interviewers also signed each form.'
Also, relation of the researchers (especially in terms of the links with the wider study within which this research is based) needs to be specified	The data on facility delivery came from a larger study exploring a variety of maternal and newborn care behaviours. Data on facility delivery were collected within the same semi-structured guide, and by the same data collectors, as the data on the other behaviours. We have clarified this in the text, and the first line of the paragraph beneath Table 1 now reads: 'Data were collected as part of a study to understand how HEWs influence maternal and newborn care behaviors, of which facility delivery was one'.
Characteristics of the respondents need to be presented in more detail – at present the text states that recruitment was done in a way to allow for breadth of characteristics but no detail is provided in terms of the outcome of the strategy	We have now included Table 3 outlining the characteristics of the IDI and narrative women, and a paragraph at the start of the results referencing the table and describing the FGD participants.
Please explain why the qualitative methods applied were selected and what the difference between the different types of data collection methods (in-depth interviews, narrative interviews and focus group discussions) was and why all of them were deployed.	We already mention that this was to allow for data triangulation, and have now added more detail in the paragraph following Table 1: 'Narrative interviews with mothers were used to capture personal experiences, in-depth interviews to capture perceptions of what was commonly done in the community, and focus group discussions to collect data that we felt would benefit from being discussed in a group interaction'.
Regarding respondents, more detail on how they were selected is required especially in light of the acknowledged limitation of potential bias stemming from the recruitment and pool of people involved in the selection The breadth of respondents – reported as a recruitment strategy – requires substantiating; would the mothers, fathers and grandmothers present the same households? If so, that considerably limits the breadth of responses and needs to be acknowledged; similarly, would snowballing be done via health providers who were also respondents? That again shrinks the pool of responses voiced by those who participated in the research. Overall, it would be helpful to know how many stems for the recruitment (with for gate keepers and	Respondents were from different families and snowball sampling was done from community respondents. In the first paragraph on page 7 we have added: 'Mothers, grandmothers and fathers, from different households, were identified either by the HEW/HDA, or through snowball sampling from the community respondents'. In the first paragraph on page 7 we have clarified that the majority of respondents were identified by the HEW/HDA. 'Mothers, grandmothers and fathers, from different households, were identified either by the

snowballing) were used and in case that number is low, to acknowledge and explain that under limitations	HEW/HDA, or through snowball sampling – with the first method providing the majority of respondents’ We have also highlighted this as a limitation more strongly in the discussion where we have added the following to the second to last paragraph on page 15: ‘In addition, those respondents identified by the HEW/HDA may have been selected because of their positive attitudes and experiences. To try to reduce this we utilized snowball sampling to identify respondents, but the majority of the respondents were identified through the HEWs/HDAs. As a result, the study respondents may have had different attitudes and experiences to families that were less favored by the HEW/HDA.
Definitions of terms such as ‘recent mothers’ (which presumably also expected to the fathers and grandmothers), ‘reasonably accessible’ and ‘reasonably functional’, ‘no unusual characteristics’ need to be specified.	Recent mothers, fathers and grandmothers: We have added on page 4: ‘All community respondents had children or grandchildren under 12 months of age, with narrative mothers having children less than three months of age to facilitate recall’. Reasonable access, no unusual characteristics and reasonably functioning have been described in the second paragraph of the methods on page 4: ‘Data were collected from areas where ‘The Last Ten Kilometers’ (L10K) programme was active in supporting the Health Extension Program. Kebeles were selected from a list, provided by L10K project staff, of kebeles considered to have a reasonably functioning HEW system, that is that they had HEWs in place that were considered to be active and working well. Other selection criteria were that the kebeles were seen as typical of the district (Woreda) with no unusual characteristics such as having a large hospital or a large industry close by, and were less than half an hours walk from a motorable road so that the study team could feasibly access them.’
Are grandmothers the mothers of the mother or the father? Would it make any difference within the cultural context studied?	We did not set any eligibility criteria for this as both maternal and paternal grandmothers can be important. The sample was predominantly paternal grandmothers, as mothers move to their husbands compound, but included a few maternal grandmothers. The grandmother FGDs were about attitudes and perceptions rather than their lived

	experiences. We have added that the grandmothers were mostly paternal in the paragraph above Table 3 on page 9.
Some of the findings seem to lack depth and offer only a limited insight into the situation, even if the conclusions appear to suggest those insights were found in the data – please amend that to allow the readers to benefit from the work done. The flow seems quite unstructured and therefore should be tightened up and reorganised to provide a clearer narrative which is easier to understand. Applying a similar structure for the individual headers such as enables and then challenges, presenting the perspectives of the different groups of respondents or other differences observed in the study to help to bring out the necessary nuances which will make the study presentations stronger and more informative. One way would be to make the key statements then supported by evidence rather than leaving the narrative to flow and lead to a conclusion which is at times mentioned and at other times implied only.	In the results we have clarified where findings were from all respondent groups to add more clarity around themes. We have added more details where we felt that it added to the readers understanding of the key findings. For each section now reported enablers first, followed by barriers where there were any reported.
More specifically, for the challenges identified or in fact encountered by the respondent, would the authors have any evidence on how they were overcome or could be overcome according to the respondents? As mentioned above with regard to transport, lack of ambulances was noted as a key challenge, yet all the respondents represented families where babies were delivered in facilities, so a method for overcoming the barrier must have been found.	See above in relation to the main challenge of transport.
Discussion: Useful insights are presented, though relatively limited reflection in terms of placing the current research in the context of the wider body of knowledge – what does the research offer that is unique and adds to the body of knowledge? How much of the work is generalisable? What are the next steps? Also, some points raised in the discussion do not appear in the Results – please amend that.	We now start the discussion with a reflection on what other qualitative studies have found about the barriers and drivers to facility delivery and what this study adds that is new: ‘A systematic review of qualitative studies exploring facility delivery classified the findings based on the quality and coherence of studies [32]. Barriers to facility delivery in which there was high confidence were: cultural barriers, such as perceptions of birth as a natural event; decision making barriers, including the role of elder women; proximity, access and cost barriers; a reliance on TBAs; and barriers related to perceived poor quality of care and mistreatment by health workers. High confidence facilitators were valuing facilities for complications and perceiving them as providing high quality of care. Previous birth experiences were both a barrier and a facilitator. Previous studies specific to Ethiopia identified similar barriers and [13-20]. In our study none of these barriers were reported, with

	the exception of accessibility issues for more remote villages. Our findings suggest that these barriers were overcome through a combination of saturation of messages around facility delivery from trusted sources, reduction in access issues, the prohibition of TBAs, power shifts away from grandmothers and positive experiences. The focus of this paper on what has driven the change process adds new insights to the literature, which to date has focused on barriers and facilitators to uptake rather than mechanisms of change. It is widely recognized that comprehensive efforts, at multiple levels, are required to successfully increase facility delivery rates [32], this is what has occurred in the study sites. Previous interventions in Ethiopia, that have focused on access barriers at one level have not been successful [29]’. We have modified the text where points raised in the discussion were not in the results. Under the new section on data quality and limitations we discuss transferability, a term we prefer to generalizability for qualitative studies. We have added more detail to this section: ‘We took several steps to maximize data quality, and took measures to improve the transferability of our findings including: using multiple study sites, purposive sampling to saturation, reflexivity, triangulation of methods and respondent groups and within and cross case analysis [48 49]. Despite this the findings may not apply to other areas with significantly different contextual issues. For example the study sites were all reasonably accessible and had reasonably functioning HEW systems. It is likely that distance and accessibility are the main factors influencing delivery location in less accessible areas, with our respondents reporting that they knew of areas where women were unable to deliver in facilities because of distance. Studies in other settings in Ethiopia would further enhance transferability, however the study findings suggest several issues that could be considered when exploring issues related to facility delivery coverage and the effectiveness of interventions to increase facility delivery rates in other settings’.
Parts of the paper seem to read as too informal, especially the summary table of the topic guide context, as well as the methods section – please revise.	The language has been revised to make it more formal.

Where quotes are used there is no punctuation (such as a colon) to introduce it and the only way of identifying a quote is by the text appearing in quotation marks and in Italics – this is not sufficient.	Colons have been added in front of all quotes.
Use of tenses should be standardised – either present or past/reported speech to be applied to the results.	We have gone through the results and ensure we use the past/reported speech throughout.
Some of the quotes are not grammatically correct and may benefit from editing – ensuring the meaning is not affected	We have gone through the quotes and corrected punctuation and grammar, while keeping the quotes as close to the original transcript as possible.
At times quotes are not self-explanatory and therefore more context is required (e.g. p. 8 ‘they told me not to deliver at home – who are ‘they’?’).	We have gone through all the quotes and inserted who ‘they’ corresponds to in [...].
Language revision suggestion: statement on p. 11 ‘their views can change given the right circumstances’ – use of the word ‘right’ gives the sentence a normative tone and should be revised to sound more neutral.	The term ‘right’ has been replaced with ‘some’. The text now reads ‘their views can change rapidly in some circumstances.’
Bibliography – please review to ensure spellings and use of low/upper case letters is consistent	We have checked all the references and the cases. In doing this we found some other areas which we have also corrected

VERSION 2 – REVIEW

REVIEWER	Elizabeth Kaselitz University of Michigan, United States
REVIEW RETURNED	02-Mar-2019

GENERAL COMMENTS	Recommend this paper for publication.
---------------------------------------

REVIEWER	Sarah Rudrum Acadia University, Canada
REVIEW RETURNED	07-Feb-2019

GENERAL COMMENTS	I have reviewed the revised manuscript, and the authors have thoughtfully revised according to reviewer feedback.
---

REVIEWER	Barbara Madaj Liverpool School of Tropical Medicine
REVIEW RETURNED	21-Feb-2019

GENERAL COMMENTS	Thank you for submitting the revised manuscript and the changes made based on the reviewers' suggestions - the main points for clarification and amendments have been addressed well. I have two points to raise with the authors:  - I apologise for the typographical error which made one of my previous comments unclear. The request for an amended holds, so I would appreciate if the authors could address my suggestion to include more detail on the role of the Health Extension Workers
---

	and particularly any changes to their roles which would help to explain and/or contextualise the research presented in the paper. This is the previous comment with the necessary correction: 'Similarly, the roles of HEWs (not CWS) need to be better explained - some information is provided already but it is not sufficient for readers to understand their role and especially the recent changes which then help to explain the findings of the research presented'. I hope this is now clearer. - The manuscript would benefit from further proofreading to eliminate small errors (e.g. Should kebele be spelt with upper or lower case letter?; HEWs not HEWS, HDAs not HDAS). I have no further queries or issues with the manuscript.
--	---

VERSION 2 – AUTHOR RESPONSE

Thanks to the reviewers for their encouragement and further useful comments.

We have checked that HEWs, HDAs and kebele are now consistent in the use of capitals.

We have also added to the section in the introduction on HEWs- this includes adding the length of their training, the specific safe motherhood role they have and the change in their role from assisting delivery at the health post to facilitating women reaching the health centre or hospital for delivery.

VERSION 3 - REVIEW

REVIEWER	Barbara Madaj Liverpool School of Tropical Medicine, UK
REVIEW RETURNED	22-Mar-2019

GENERAL COMMENTS	All clarifications and recommendations have been addressed. I have no further comments on the manuscript
--